# Longitudinal mediation analysis of the factors associated with trajectories of posttraumatic stress disorder symptoms among postpartum women in Northwest Ethiopia: Application of the Karlson-Holm-Breen (KHB) method

**Marelign Tilahun Malaju** [1,2]*, **Getu Degu Alene**[2], **Telake Azale Bisetegn**[3]

**1** Department of Public Health, College of Health Sciences, Debre Tabor University, Debre Tabor, Ethiopia, **2** School of Public Health, College of Medicine and Health Sciences, Bahir Dar University, Bahir Dar, Ethiopia, **3** School of Public Health, College of Medicine and Health Sciences, University of Gondar, Gondar, Ethiopia

* marikum74@gmail.com

## Abstract

### Introduction

In recent years, literatures identified childbirth as a potentially traumatic experience resulting in posttraumatic stress disorder (PTSD), with 19.7 to 45.5% of women perceiving their child-birth as traumatic. A substantial variation in PTSD symptoms has been also indicated among women who experience a traumatic childbirth. However, there has been no research that has systematically investigated these patterns and their underlying determinants in postpartum women in Ethiopia.

### Objective

The aim of this study was to investigate the trajectories of PTSD symptoms and mediating relationships of variables associated with it among postpartum women in Northwest Ethiopia.

### Methods

A total of 775 women were recruited after childbirth and were followed at the 6th, 12th and 18th week of postpartum period during October, 2020 –March, 2021. A group-based trajectory modeling and mediation analysis using KHB method were carried out using Stata version 16 software in order to determine the trajectories of PTSD symptoms and mediation percentage of each mediator on the trajectories of PTSD symptoms.

### Results

Four distinct trajectories of postpartum posttraumatic stress disorder symptoms were identified. Perceived traumatic childbirth, fear of childbirth, depression, anxiety, psychological

**Data Availability Statement:** All relevant data are within the manuscript and its Supporting Information files

**Funding:** The author(s) received no specific funding for this work.

**Competing interests:** The authors have declared that no competing interests exist.

violence, higher WHODAS 2.0 total score, multigravidity, stressful life events of health risk, relational problems and income instability were found to be predictors of PTSD with recovery and chronic PTSD trajectory group membership. Depression and anxiety not only were strongly related to trajectories of PTSD symptoms directly but also mediated much of the effect of the other factors on trajectories of PTSD symptoms. In contrast, multiparity and higher mental quality of life scores were protective of belonging to the PTSD with recovery and chronic PTSD trajectory group membership.

## Conclusion

Women with symptoms of depression, anxiety, fear of childbirth and perceived traumatic childbirth were at increased risk of belonging to recovered and chronic PTSD trajectories. Postnatal screening and treatment of depression and anxiety may contribute to decrease PTSD symptoms of women in the postpartum period. Providing adequate information about birth procedures and response to mothers' needs during childbirth and training of health care providers to be mindful of factors that contribute to negative appraisals of childbirth are essential to reduce fear of childbirth and traumatic childbirth so as to prevent PTSD symptoms in the postpartum period.

## Introduction

There is a growing body of literature which identified childbirth as a potentially traumatic experience resulting in posttraumatic stress disorder (PTSD), with 19.7 to 45.5% of women perceiving their childbirth as traumatic [1–3]. Childbirth differs from other traumatic experiences in that it is predictable and, for many women, a pleasurable experience [4]. For other women, however, it has been suggested that childbirth is a psychologically stressful event that meets the PTSD trauma criteria, as the DSM-5 includes PTSD under trauma and stressor-related disorders [1,4–8]. Traumatic birth has been defined as an event occurring during labour and birth that may be a serious threat to the life and safety of the mother and/or child [9]. Post-traumatic stress disorder refers to a cluster of psychological symptoms that develop following exposure to a severe stressor or traumatic event associated with a real or perceived threat of death or threat to physical integrity of the person or others. It is characterized by symptoms of persistent experiencing the event, avoidance of stimuli associated with the event, negative mood alterations, and increased arousal and reactivity [7].

Studies have reported 3–20% prevalence rates of post-traumatic stress disorder in the postpartum period [10–14]. In a meta-analysis study, it has been reported that 4% of women were found to have PTSD symptoms in the postpartum period and this increases to 18.5% in women with obstetric complications [13]. A study in Norway, reported that 1.8% of women had PTSD following childbirth [15], whereas in Iran it has been found that 20% of women had PTSD in the postpartum period [14]. A number of studies have also indicated that PTSD can have a negative impact on the psychological wellbeing of mothers, their relationships with their infant and husband/partner and birth outcomes [16–19]. There are also indications that it may affect infant emotion regulation and development [20,21].

PTSD is not the result of a single cause (i.e., a traumatic stressor), but is the consequence of various interacting variables [3]. According to the diathesis-stress model of birth-related PTSD, it is the outcome of the interplay between antepartum vulnerability factors, the risk

factors during delivery and maintaining postnatal factors that interact over time during the perinatal period [11]. In a meta-analysis [11], vulnerability factors such as depression during pregnancy, fear of childbirth, poor health or complications in pregnancy, a history of PTSD and previous counselling were found to be associated with postpartum PTSD symptoms. In terms of risk factors, subjective birth experience (including lack of control during delivery), operative birth, lack of support from staff and dissociation were reported as predictors of PTSD symptoms in the postpartum period. With regard to maintaining factors, poor coping strategies, higher levels of stress and higher levels of depression were found to be predicters of postpartum PTSD symptoms this study [11]. Other meta-analysis studies have also provided an overview of risk factors for developing postpartum PTSD symptoms. These include current levels of depression, trait anxiety, labour experiences and obstetric procedures, previous traumatic experiences, history of psychopathology, negative aspects in staff and mother contact, feelings of lack of control over the delivery situation and lack of support from the partner [10,22].

PTSD following childbirth usually occurs as a result of complications during pregnancy or childbirth (10, 22). However, it may also be a continuation of pre-existing PTSD, a reactivation of PTSD triggered by childbirth related events that had previously resolved or new-onset PTSD in response to an event which is not related to childbirth [2,23,24]. Medical complications or interventions, such as emergency caesarean section, preeclampsia and women's subjective experience of birth as negative and traumatic are associated with PTSD symptoms [25–32]. Despite the fact that medical interventions and complications such as instrumental deliveries, emergency caesarean sections, or preeclampsia increase the risk of a traumatic birth experience [1,25], a childbirth without complications can also be traumatic due to a loss of control, a perceived threat, or physical harm to the mother or baby [1,33,34]. Fear of childbirth also puts a woman at greater risk of developing PTSD in response to the subsequent birth [11]. Symptoms of depression and anxiety, are also significantly associated with PTSD following childbirth [10,11]. Additionally, low social support was found to be associated with development of PTSD during the postpartum period [25,26,35].

There has also been evidence of a significant difference in PTSD symptoms among women who have had a traumatic birth, according to studies [1]. Some of the limited studies that have been done show a decrease in the prevalence of PTSD over time, indicating that some women recover during the initial months after giving birth [3]. Other studies have documented the persistence or aggravation of PTSD symptoms over time and have identified chronic PTSD [5,36]. Following a difficult birth, new incidences of PTSD have also been documented [5]. As a result, PTSD is a very heterogeneous construct, and women's reactions and response patterns after a traumatic birth vary greatly [1].

Despite the various trajectories that PTSD symptoms can take after childbirth [1], it is understudied [1], and no research has looked into these patterns and their underlying factors in Ethiopian postpartum women. A thorough analysis of these trajectories in postpartum women allows for a more comprehensive knowledge of both favorable and negative outcomes following a traumatic birth [1]. Identifying subgroups of women after traumatic birth and predictors of such trajectory groups may improve understanding of the development, course, and outcomes of birth-related PTSD, as well as facilitate differentiation between low- and high-risk women for postpartum PTSD, which could inform appropriate postpartum PTSD prevention and treatment strategies [1].

Furthermore, a vacuum in the research has been noted due to a dearth of studies investigating the mediating links between susceptibility and risk variables in order to better understand the development of postpartum PTSD symptoms [4,11]. Because the majority of studies have focused on the prevalence of postpartum PTSD symptoms and the factors that influence them,

the mediating relationships between variables associated with PTSD symptom trajectories have been overlooked [4]. Because of this vacuum in the research, the current study looked into how anxiety and depression interacted with traumatic childbirth, fear of childbirth, health risk, and functional status in predicting postpartum PTSD trajectories. The selection of these specific variables was based on the report of a previous research that linked anxiety and depression to traumatic childbirth, fear of childbirth, health risk, functional status and postpartum PTSD symptoms [3,6,37].

Furthermore, we wanted to see the possible linkage of anxiety and depression symptoms with traumatic childbirth, fear of childbirth, health risk and functional status, taking into consideration how they interacted with other risk variables [11]. The investigation of mediating links between susceptibility and risk factors for developing postpartum PTSD symptoms is crucial because it enables for the prevention of PTSD symptoms by identifying women who are at higher risk and providing them with adequate postpartum care [3,4]. Therefore, the aim of this study was to investigate the trajectories of PTSD symptoms and mediating relationships among the variables associated with it during the postpartum period in Northwest Ethiopia.

## Methods and materials

### Study design and study area

Data used for this study were collected as part of the health facility linked community based prospective follow-up study conducted in Northwest Ethiopia to determine the effect of maternal morbidities on maternal health related quality of life, functional status and mental health problems. The details of the methods were described elsewhere [38]. In brief, postpartum women were recruited in four hospitals of south Gondar zone, Northwest Ethiopia. The data collection took place between October 1, 2020 and March 30, 2021. South Gondar is located at 650 km Northwest from Addis Ababa the capital city of Ethiopia.

### Study population

A total of 775 women consented to participate in the study and participated at the first, second and third follow-up of the study (6th, 12th and 18th week of postpartum period). Recruitment of the study participants was done after child birth and before the time of discharge. Women with any of the direct and indirect maternal morbidities were recruited into the exposed group and those without the direct and indirect maternal morbidities were into the non-exposed group based on the WHO maternal morbidity criteria [39].

### Eligibility/Inclusion criteria

Women aged 15 years and above, with preterm, term or post term delivery and with live birth, still birth or fetal death were included in the study. The PTSD criterion A was not considered as an exclusion criterion, because childbirth related negative events and emotions that do not satisfy the criterion A can cause symptoms that could qualify as a PTSD diagnosis [3].

### Sampling procedure

All exposed women with direct maternal morbidity included in the study and non-exposed women without direct maternal morbidities were selected by simple random sampling method using their chart number on daily bases. With 1:2 ratio of exposed to non-exposed mothers, this recruitment procedure continued prospectively until the required sample size was fulfilled. Women were asked for consent to participate in the study and after getting their consent and full address, appointments were made at their home to collect the data for the follow up study.

**Dependent variable.**   Trajectories of posttraumatic stress disorder was taken as the outcome variable.

**Independent variables.**   Direct maternal morbidities(obstetric hemorrhage, hypertensive disorders, obstructed labour, puerperal sepsis, gestational diabetes mellitus, perineal tear), indirect maternal morbidities (anemia, malaria, hypertension, asthma, tuberculosis, HIV), socio-demographic variables (age, educational status, marital status, religion, ethnicity, occupation, monthly expenditure), residence, obstetric variables (parity, mode of delivery, gestational age at birth, birth weight, birth interval, fetal death, unwanted pregnancies, antenatal care visit, history of abortion), health related quality of life and psychosocial factors (social support and fear of child birth) were taken as the main independent variables.

**Mediator variables.**   functional status, depression and anxiety were taken as mediators of the independent variables associated with the outcome variable.

## Measures of variables

**Depression, anxiety and stress.**   The short version of depression, anxiety and stress scale 21 (DASS-21) questionnaire was used to measure depression, anxiety and stress. The instrument has 21 items with three domains. Each domain comprises seven items assessing symptoms of depression, anxiety and stress. In this study a score $\geq 10$ was considered for a mother to have a symptom of depression. A cut-off score of $\geq 8$ was considered to have symptoms of anxiety and a score of $\geq 15$ was considered to have symptoms of stress. This instrument was validated and used previously in Ethiopia [40,41].

**Posttraumatic stress disorder.**   The childbirth stressor was operationalized by using the Traumatic Event Scale (TES) [42,43]. In this scale, the items concerning criterion A (stressor) were formulated as follows:

1. "The childbirth was a trying experience."

2. "The childbirth was a threat to my physical integrity."

3. "During the childbirth I was afraid that I was going to die."

4. "During the childbirth I felt anxious/helpless/horrified."

Four alternative answers follow each statement: "not at all," "somehow," "much," and "very much." Criterion A is fulfilled if either of the alternatives "much" or "very much" on item 1, 2 and/or 3, and 4 is marked [42,43].

After the questions regarding criterion A, we have used the Posttraumatic Stress Disorder Checklist for DSM-5 (PCL-5) comprising the 20 PTSD symptoms (criterion B, C and D) to measure PTSD. The instrument contains 20 items, including three new PTSD symptoms (compared with the PTSD Checklist for DSM-IV): blame, negative emotions and reckless or self-destructive behavior [44]. A total-symptom score of zero to 80 can be obtained by summing the items. A score of 31–33 is optimal to determine PTSD symptoms and a score of $\geq 33$ is recommended when further psychometric testing is not available [45,46]. Therefore, a score of $\geq 33$ was considered to have symptoms of PTSD for this study. The instrument was used previously in Ethiopia [46].

## Health related quality of life

The quality of life was measured by the Amharic version of the World Health Organization Quality of Life (WHOQOL-BREF) instrument [47]. It consists of 24 items to measure perception of quality of life in four domains, including physical health, psychological, social relationships and environment, and two items on overall QOL and general health. The domain scores

were transformed into a linear scale between 0 and 100 following the scoring guidelines [47]. A higher score indicated a better quality of life. The WHOQOL-BREF has been previously validated and used in Ethiopia [48].

**Functional status.**    To measure maternal functional status, the 36-item form of the WHO-DAS 2.0 instrument was used (the 32-item form was used for participants who were unemployed and no longer in school). The WHODAS has been previously validated and used in Ethiopia [49–52]. The WHODAS 2.0 is designed to measure activity functioning and participation in daily living activities in the previous 30 days. The instrument provides a common way of measuring the impact of any health condition in terms of functioning. It is not targeted to a specific disease, so it can be used to compare disability due to different conditions. The WHODA 2.0 consists of six domains: cognition (understanding and communication), mobility, self-care, getting along with people, life activities and participation in society. Results provided a profile of functioning within the domains as well overall score. Total WHODAS 2.0 scores can range from zero to 100, with higher numbers indicating greater impairment of day-to-day functioning [53].

**Fear of child birth.**    The Wijma Delivery Expectation/Experience Questionnaire (W-DEQ) was used to measure fear of child birth. The W-DEQ has been designed specially to measure fear of child birth operationalized by the cognitive appraisal of the delivery. This 33-item rating scale has a 6-point Likert scale as a response format, ranging from ' not at all' (= 0) to ' extremely' (= 5), yielding a score-range between 0 and 165. The Internal consistency and split-half reliability of the W-DEQ was checked in previous studies in Ethiopia with the Cronbach's alpha score of 0.932 (54, 55). A score of $\geq$ 85 was considered to have fear of child birth for this study [54,55].

## Social support

The Oslo 3-items social support scale with scores ranging from 3 to 14 was used to measure social support. The social support scores were categorized into poor or no social support for scores less than nine. Scores between 9 and 14 were considered moderate to strong support and merged together as "yes" for social support. The Oslo 3-items social support scale was validated and previously used in Ethiopia [56–58].

**Stressful life events.**    The List of Threatening Experiences (LTE) was used to measure experience of stressful life events during the six months period [59]. The 12 items were categorized into five categories namely health risks, loss of a loved one, relationship difficulties, income instability and legal problems [60]. The list of threatening experiences (LTE-12) has been used in a population level study in Ethiopia [61,62].

## Domestic violence

Domestic violence was measured by the WHO (2005) multi country study questionnaire. This questionnaire has four items for psychological violence, six items for physical violence and three items for sexual violence [63].

**Data collection and quality control.**    Administering baseline questionnaire and diagnosis of direct and indirect maternal morbidities using the WHO maternal morbidity working group criteria, were done by health professionals working in the Gynecology and Obstetrics wards of the study Hospitals. The questionnaire consisted of a patient interview and record review. The interview was on socioeconomic status, medical and obstetric history and clinical symptoms. The record review was intended to extract information on selected laboratory tests and results for hemoglobin, HIV, malaria (rapid diagnostic test or smear) and glucometer (random blood sugar). The DASS-21 and PCL-5 were administered by health extension

workers at the first, second and third home visit (6th, 12th and 18th week of postpartum period). Training was given for data collectors and supervision was done by the principal investigator. During the training process, data collectors carefully reviewed each question and pretest was done before the study commences. The investigator and data collectors have checked the final version of the questionnaire and update as required based on the pretest.

**Data processing and analysis.** Trajectories of PTSD symptoms were determined by group-based trajectory modeling using the Traj package of Stata 16. The group-based trajectory modeling was used to identify distinct homogenous clusters of postpartum women with similar progressions of PTSD symptoms during the follow up period [64]. A distinct trajectory consists of a group of individuals who share a common underlying pattern of PTSD symptoms change over time [64]. Censored normal finite mixture model was used to estimate trajectories of PTSD symptoms over the postpartum period (at 6th, 12th and 18th week of postpartum period).

The identification process of appropriate group trajectories was based on the selection and reporting procedures outlined by Nagin et al [65]. In the model selection process, the Bayesian Information Criterion (BIC) was used to determine the best model underlying the group selection and functional form. The closer the negative BIC value is to zero, the better is the fit of the model. A difference in the BIC value of at least 10 points between two models indicates that the model with the lower BIC value has a better model fit [65].

We have also assessed the posterior probabilities of group membership and required that average posterior probability reached 0.70 or higher to be a distinct classification group. Entropy, a statistic that ranges from 0.00 to 1.00 which is a summary indicator of the conditional probabilities of individuals' group membership has been also used. High values of entropy ($> .80$) indicate that individuals are classified with confidence (i.e., the model is generally pretty sure that persons belong to a particular class) and there is adequate separation between the latent classes [66]. Additionally, we required at least 5% of the sample to be present in a particular group and all trajectories were distinct from one another by visual assessment of trajectory figures looking for nonoverlapping confidence intervals [65,67,68]. The labeling of each trajectory was based on a previous research work [1].

Then, multinomial logistic regression model was carried out to identify factors that were associated with probability of group (trajectory) membership. First, unadjusted bivariable associations between each predictor and trajectory membership were tested in order to identify predictors having a p-value of $\leq 0.2$ to enter into the multivariable multinomial logistic regression. Next, multivariable multinomial logistic regression model was fitted to determine factors associated with membership to PTSD symptom trajectories. We report the odds ratio of group membership with 95% CI and a p-value of $\leq 0.05$ for statistical significance.

Prior to fitting the Structural Equation Model (SEM), a confirmatory factor analysis (CFA) was conducted to test the model fit of the Posttraumatic Stress Disorder Checklist for DSM-5 (PCL-5) scores. We estimated the model fitness by using the comparative fit index (CFI), Tucker-Lewis Index (TLI) and the root-mean-square error of approximation (RMSEA). Both the TLI and CFI should be greater than 0.90, but the RMSEA value should be less than 0.08 to judge the model as reasonably fitting the data [3,69]. The direct, indirect, and total effects of independent variables were reported in the form of standardized beta coefficients. Estimated effects for which $p < 0.05$ were considered as being statistically significant.

Finally, the direct and indirect effects of independent variables on the postpartum PTSD trajectories were assessed using the recently developed Karlson-Holm-Breen (KHB) method, which is appropriate for mediation analyses in nonlinear models [70,71]. The KHB decomposition method has been found to keep the features of decomposing a linear model [72] and permits a vector of mediators to be analyzed even in a nonlinear model [71]. This method allows

researchers to examine multiple mediators simultaneously [73]. It also enables to decompose the total effect of a key independent variable in a logistic regression model into the sum of the direct and indirect effects. The KHB method also estimates all effects (i.e., overall, direct, and indirect) on the same scale, making comparisons across different mediators or coefficients reliable. It also allows for the calculation of the mediated percentage, which is interpreted as the percentage of the main association that can be explained by the mediator. The mediated percentage was only considered significant when the total and indirect effects were significant [74]. Moreover, the KHB method allows researchers to include other confounding variables (as concomitants) into the models without the scale identification issue to control the decomposition of any potential confounding factors. The KHB method was implemented by a user-written KHB command in Stata 16.0. which applies decomposition properties of linear models to the logit model [71].

**Ethical considerations.**   This study was approved by the institutional review board of Bahir Dar University. Each study participant has given written informed consent before participating in the study. Assent was also obtained from teenage mothers whose age is less than 18 years, in addition to informed consent from their care givers. Using codes, passwords and limiting access to the data only for the investigators were the measures taken to ensure the confidentiality of the data. Data collectors read out and assisted participants to fill out the consent form if participants were unable to read and write.

## Results

At baseline 779 women were recruited and 775(99.5%) of them participated at the first, second and third follow-up of the study (6th, 12th and 18th week of postpartum period). Four mothers were lost to follow up because of changing their place of living and going out of the study area. The mean age of the study participants was 26.32(±4.38). Almost all of them 774(99.9%) were Amhara by ethnicity and 742(95.7%) were followers of Orthodox Christianity. Other socio-demographic characteristics of mothers are shown in Table 1.

### Confirmatory factor analysis for PCL-5

The model for postnatal posttraumatic stress disorder had a satisfactory fit according to the various fit indices (CFI = 0.83, TLI = 0.82 and SRMR = 0.048) and all the factor loadings were significant at p < 0.001. The estimated standardized path loadings for the structural equation model are shown in Fig 1.

### Identification of the PTSD trajectories

As indicated in Table 2, the group-based trajectory analysis indicates that four latent trajectory class best fitted the PCL-5 data. The four trajectories of PTSD symptoms were labeled as resilient with mean of zero, vulnerable with mean of < 33 cut-off, PTSD with recovery and Chronic PTSD.

The resilient group with 66.8% of the total cohort, had consistently (stable) no PTSD symptom patterns throughout the follow up period. The vulnerable group (21.3% of the cohort) consisted of postpartum women with PTSD symptoms less than the PCL-5 cut-off level (33). This trajectory group exhibited decreasing patterns (recovery) over the follow up period. The third trajectory, labeled as PTSD with recovery group consisted of 5.3% of postpartum women with PTSD symptoms above the PCL-5 cut-off level (33). This trajectory group also exhibited decreasing patterns (recovery) over the follow up period. However, the fourth trajectory of the PTSD symptoms labeled as chronic PTSD group, exhibited relatively stable patterns of higher

**Table 1. Socio-demographic characteristics of postpartum women by PTSD trajectory groups in Northwest Ethiopia, 2021.**

| Variables | Trajectory group | | | | Total n (%) |
|---|---|---|---|---|---|
| | Resilient with mean of zero n (%) | Vulnerable with mean of < 33 cut-off) n (%) | PTSD with recovery n (%) | Chronic PTSD n (%) | |
| **Age = Mean(±SD)** | 26.41 (±4.25) | 26.08 (±4.42) | 25.93 (±4.12) | 26.48 (±5.69) | 26.32 (±4.38) |
| **Age category** | | | | | |
| < 18 years | 0 (0.0) | 0 (0.0) | 1 (0.1) | 1 (0.1) | 2 (0.3) |
| 18–25 years | 236 (30.5) | 86 (11.1) | 19 (2.5) | 20 (2.6) | 361 (46.6 |
| 26–35 years | 266 (34.3) | 70 (9.0) | 20 (2.6) | 27 (3.5) | 383 (49.4) |
| 36–45 | 19 (2.5) | 7 (0.9) | 1 (0.1) | 2 (0.3) | 29 (3.7) |
| **Residence** | | | | | |
| Urban | 520 (67.1) | 162 (20.9) | 41 (5.3) | 48 (6.2) | 771 (99.5) |
| Rural | 1(0.1) | 1(0.1) | 0(0.0) | 2 (0.3) | 4(0.5) |
| **Ethnicity** | | | | | |
| Amhara | 521 (67.2) | 163 (21.0) | 41 (5.3) | 49 (6.3) | 774 (99.9) |
| Tigre | 0(0.0) | 0(0.0) | 0 (0.0) | 1 (0.0) | 1(0.1) |
| **Religion** | | | | | |
| Orthodox | 509 (65.7) | 154 (19.9) | 35 (4.5) | 44 (5.7) | 742 (95.7) |
| Muslim | 10 (1.3) | 9 (1.2) | 5 (0.6) | 6 (0.8) | 30 (3.9) |
| Protestant | 2 (0.3) | 0 (0.0) | 1 (0.1) | 0 (0.0) | 3 (0.4) |
| **Education status** | | | | | |
| Read and write | 42 (5.4) | 16 (2.1) | 3 (0.4) | 4 (0.5) | 65 (8.4) |
| Grade 1–8 (Primary School) | 91 (11.7) | 32 (4.1) | 5 (0.6) | 8 (1.0) | 136 (17.5) |
| Grade 9–12 (secondar School) | 146 (18.8) | 46 (5.9) | 10 (1.3) | 17 (2.2) | 219 (28.3) |
| Certificate/Diploma | 149 (19.2) | 40 (5.2) | 15 (1.9) | 13 (1.7) | 217 (28.0) |
| Degree and higher | 93 (12.0) | 29 (3.7) | 8 (1.0) | 8 (1.0) | 138 (17.8) |
| **Occupation** | | | | | |
| Gov't employed | 150 (19.4) | 46 (5.9) | 17 (2.2) | 17 (2.2) | 230 (29.7) |
| Merchant/Student | 99 (12.8) | 29 (3.7) | 7 (0.9) | 10 (1.3) | 145 (18.7) |
| Housewife | 248 (32.0) | 81 (10.5) | 16 (2.1) | 22 (2.8) | 367 (47.4) |
| Farmer/Daily laborer | 24 (3.1) | 7 (0.9) | 1 (0.1) | 1 (0.1) | 33 (4.3) |
| **Marital Status** | | | | | |
| Married | 515 (66.5) | 160 (20.6) | 41 (5.3) | 46 (5.9) | 762 (98.3) |
| Single/widowed/divorced | 6 (0.8) | 3 (0.4) | 0 (0.0) | 4 (0.5) | 13 (1.7) |
| **Monthly expenditure** | | | | | |
| ≤3000 Ethiopian currency | 134 (17.3) | 55 (7.1) | 7 (0.9) | 10 (1.3) | 206 (26.6) |
| 3001–4000 Ethiopian currency | 131 (16.9) | 36 (4.6) | 12 (1.5) | 13 (1.7) | 192 (24.8) |
| ≥ 4001 Ethiopian currency | 256 (33.0) | 72 (9.3) | 22 (2.8) | 27 (3.5) | 377 (48.6) |

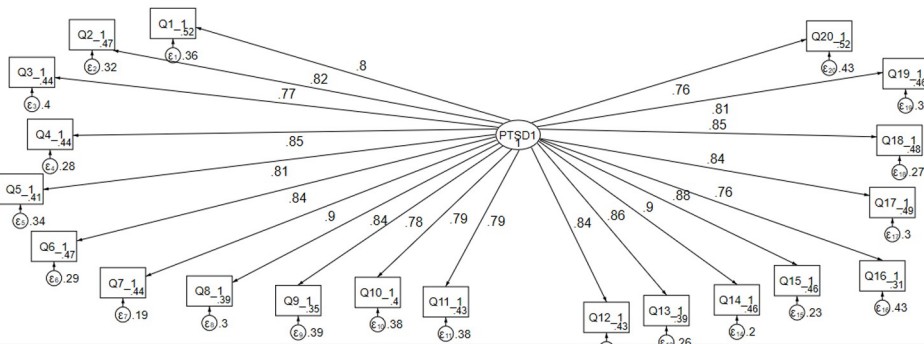

**Fig 1. Results of a standardized factor loadings of a measurement model for PTSD symptoms (N = 775), Northwest, Ethiopia, 2021.** Note: β's are standardized estimates, P-value < 0.001.

**Table 2. Fit indices for group-based trajectory of PCL-5 scores among postpartum women in Northwest Ethiopia, 2021.**

| Number of trajectories for PTSD (PCL-5) | Polynomial function order for PCL-5 scores | BIC | Posterior probability | Entropy | Proportions in each trajectory | | | |
|---|---|---|---|---|---|---|---|---|
| | | | | | 1 | 2 | 3 | 4 |
| 1 | 1 | -4161.03 | 1.0 | NA | 100 | | | |
| 2 | 1,1 | -3605.87 | 0.999 | 0.946 | 68.1 | 31.9 | | |
| 3 | 2,2,2 | -3309.87 | 0.999 | 0.969 | 66.7 | 22.5 | 10.8 | |
| **4** | **1,1,1,1** | **-3262.15** | **.977** | **0.965** | **66.8** | **21.3** | **5.4** | **6.5** |

Note: NA = Not applicable.

PTSD symptom scores (above the PCL-5 cut-off level) throughout the follow up period (see Fig 2).

## PTSD trajectory groups by obstetrics and psychosocial variables

Table 3 describes the comparison of obstetrics and psychosocial variables across trajectory groups. Out of 252(32.5%) mothers with direct maternal morbidity, 141 (18.2), 97 (12.5), 6 (0.8) and 8 (1.0) of them were in the trajectory group of resilient with mean of zero, vulnerable with mean of < 33 cut-off, PTSD with recovery and chronic PTSD respectively. Out of women with perceived traumatic birth, 156 (20.1%), 73 (9.4%), 32 (4.1%) and 45 (5.8%) of them were in the trajectory group of resilient with mean of zero, vulnerable with mean of < 33 cut-off, PTSD with recovery and chronic PTSD respectively (see Table 3).

## Predictors of PTSD (PCL-5) trajectory group membership

Predictors of the PTSD trajectory group membership which were found to be statistically significant in the multinomial logistic regression analysis were presented in Table 4.

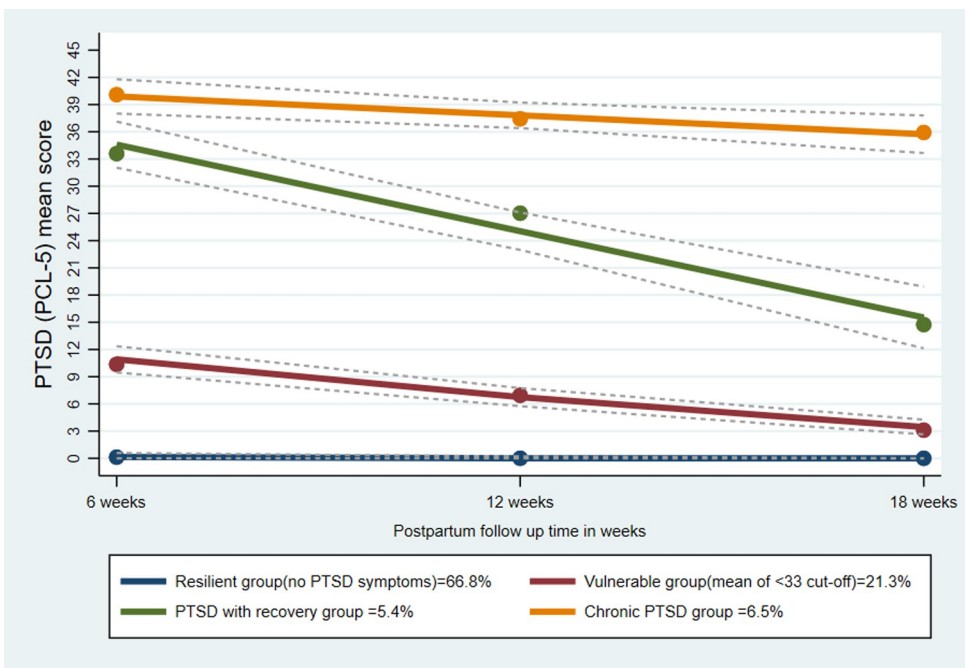

**Fig 2. Trajectories of PTSD symptoms among postpartum women, Northwest Ethiopia, 2021.**

**Table 3. Obstetrics and psychosocial variables by PTSD trajectory groups among postpartum women in Northwest Ethiopia, 2021.**

| Variables | Trajectory group | | | | Total n (%) |
|---|---|---|---|---|---|
| | Resilient with mean of zero n (%) | Vulnerable with mean of < 33 cut-off) n (%) | PTSD with recovery n (%) | Chronic PTSD n (%) | |
| **Perceived traumatic birth** | | | | | |
| Yes | 156 (20.1) | 73 (9.4) | 32 (4.1) | 45 (5.8) | 306 (39.5) |
| No | 365 (47.1) | 90 (11.6) | 9 (1.2) | 5 (0.6) | 469 (60.5) |
| **Direct maternal morbidity** | | | | | |
| Yes | 141 (18.2) | 97 (12.5) | 6 (0.8) | 8 (1.0) | 252 (32.5) |
| No | 380 (49.0) | 66 (8.5) | 35 (4.5) | 42 (5.4) | 523 (67.5) |
| **Indirect maternal morbidity** | | | | | |
| Yes | 111 (14.3) | 89 (11.5) | 5 (0.6) | 5 (0.6) | 210 (27.1) |
| No | 410 (52.9) | 74 (9.5) | 36 (4.6) | 45 (5.8) | 565 (72.9) |
| **Social support** | | | | | |
| Poor social support | 227 (29.3) | 80 (10.3) | 15 (1.9) | 23 (3.00 | 345 (44.5) |
| Strong social support | 294 (37.9) | 83 (10.7) | 26 (3.4) | 27 (3.5) | 430 (55.5) |
| **Mode of delivery** | | | | | |
| SVD/instrumental delivery | 383 (50.5) | 96 (12.6) | 31 (4.1) | 40 (5.3) | 550 (72.5) |
| Elective/emergency C/S | 122 (16.1) | 67 (8.8) | 10 (1.3) | 10 (1.3) | 209 (27.5) |
| **Fetal outcome** | | | | | |
| Live birth | 509 (65.7) | 161 (20.8) | 40 (5.2) | 49 (6.3) | 16 (2.1) |
| Still birth/IUFD/fetal anomaly | 12 (1.5) | 2 (0.3) | 1 (0.1) | 1 (0.1) | |
| **Fear of child birth** | | | | | |
| Yes | 108 (13.9) | 76 (9.8) | 32 (4.1) | 29 (3.7) | 245 (31.6) |
| No | 413 (53.3) | 87 (11.2) | 9 (1.2) | 21 (2.7) | 530 (68.4) |
| **Health risk** | | | | | |
| Yes | 27 (3.5) | 45 (5.8) | 13 (1.7) | 9 (1.2) | 94 (12.1) |
| No | 494 (63.7) | 118 (15.2) | 28 (3.6) | 41 (5.3) | 681 (87.9) |
| **Income instability** | | | | | |
| Yes | 15 (1.9) | 22 (2.8) | 19 (2.5) | 19 (2.5) | 75 (9.7) |
| No | 506 (65.3) | 141 (18.2) | 22 (2.8) | 31 (4.0) | 700 (90.3) |
| **Relational problems** | | | | | |
| Yes | 20 (2.6) | 20 (2.6) | 19 (2.5) | 22 (2.8) | 81 (10.5) |
| No | 501 (64.6) | 143 (18.5) | 22 (2.8) | 28 (3.6) | 694 (89.5) |
| **Death of loved one** | | | | | |
| Yes | 13 (1.7) | 22 (2.8) | 13 (1.7) | 11 (1.4) | 59 (7.6) |
| No | 508 (65.5) | 141 (18.2) | 28 (3.6) | 39 (5.0) | 716 (92.4) |

Note: SVD, spontaneous vaginal delivery, IUFD, intra uterine fetal death.

Women with perceived traumatic birth were 5.0 and 7.0 times more likely to belong to the PTSD with recovery and chronic PTSD trajectory groups respectively (PTSD with recovery: [OR & (95%CI) = 4.86 (1.86, 12.65)]; chronic PTSD: [OR & (95%CI) = 14.07 (4.48, 44.16)] relative to the resilient trajectory group. Women with stressful life event of health risk were also 4.0 times more likely to belong to the latent class of PTSD with recovery [OR & (95%CI) = 3.69 (1.17, 11.59)] compared with participants in the resilient trajectory group.

Women with psychological violence were almost 8.0 times more likely to belong to the chronic PTSD trajectory group (chronic PTSD: [OR & (95%CI) = 7.56 (1.14, 50.08)], compared with women in the resilient trajectory group. Similarly, participants with stressful life event of death of loved one and women with cesarean section/instrumental delivery were 2.5[OR & (95%CI) = 2.53 (1.07, 5.99)] and 2.0 times [OR & (95%CI) = 2.01 (1.10, 3.66)] more likely to belong to the PTSD vulnerable trajectory group respectively relative to the resilient trajectory group.

A one unit increase in fear of childbirth score, increased the risk of belonging to the PTSD with recovery group and chronic PTSD group by 1.03 and 1.02 times respectively (PTSD with

**Table 4. Significant predictors of PTSD trajectories among postpartum women, Northwest Ethiopia, 2021.**

| Explanatory variables | PTSD trajectories (reference group: resilient with mean of zero) | | | | | |
|---|---|---|---|---|---|---|
| | Vulnerable Vs resilient group | | PTSD with recovery Vs resilient group | | Chronic PTSD Vs resilient group | |
| | AOR (95%CI) | P-value | AOR (95%CI) | P-value | AOR (95%CI) | P-value |
| **Socioeconomic status variables:** | | | | | | |
| < 3000 Ethiopian currency | 2.16 (1.25, 3.74) | 0.006 | | | | |
| **Quality of life & functional status variables:** | | | | | | |
| Mental quality of life score | 0.96 (0.94, 0.97) | < 0.001 | 0.95 (0.92, 0.98) | 0.001 | 0.92 (0.89, 0.95) | < 0.001 |
| WHODAS 2.0 total score | 1.09 (1.07, 1.1) | < 0.001 | 1.18 (1.13, 1.23) | < 0.001 | 1.17 (1.12, 1.22) | < 0.001 |
| **Stressful life events variables:** | | | | | | |
| Relational problems: Yes | | | 9.83 (2.87, 33.61) | < 0.001 | 10.04 (2.91, 34.64) | < 0.001 |
| Income Instability: Yes | | | 7.16 (2.01, 25.46) | 0.002 | 4.04 (1.10, 14.84) | 0.036 |
| Health risk: Yes | 3.37 (1.79, 6.31) | < 0.001 | 3.69 (1.17, 11.59) | 0.025 | | |
| Death of loved one: Yes | 2.53 (1.07, 5.99) | 0.034 | | | | |
| **Obstetric and birth related variables:** | | | | | | |
| Perceived traumatic birth: Yes | | | 4.86 (1.86, 12.65) | 0.001 | 14.07 (4.48, 44.16) | < 0.001 |
| Parity | | | 0.37 (0.16, 0.85) | 0.019 | 0.22 (0.10, 0.49) | < 0.001 |
| Gravidity | | | 2.38 (1.04, 5.44) | 0.04 | 3.39 (1.56, 7.38) | 0.002 |
| Mode of birth: C/s or instrumental delivery | 2.01 (1.10, 3.66) | 0.023 | | | | |
| **Psychosocial variables:** | | | | | | |
| Depression | 1.14 (1.04, 1.26) | 0.006 | 1.32 (1.13, 1.55) | < 0.001 | 1.44 (1.22, 1.68) | < 0.001 |
| Anxiety | 1.26 (1.14, 1.39) | < 0.001 | 1.43 (1.21, 1.69) | < 0.001 | 1.38 (1.17, 1.63) | < 0.001 |
| Social support | 0.80 (0.72, 0.89) | < 0.001 | | | | |
| Fear of childbirth score | 1.01 (1.001, 1.02) | 0.025 | 1.03 (1.01, 1.06) | 0.001 | 1.02 (1.004, 1.04) | 0.018 |
| **Domestic violence variables:** | | | | | | |
| Psychological violence: Yes | | | | | 7.56 (1.14, 50.08) | 0.036 |

recovery: [OR & (95%CI) = 1.03 (1.01, 1.06)]; chronic PTSD: [OR & (95%CI) = 1.02 (1.004, 1.04)]. Similarly, a one unit increase in WHODAS 2.0 total score (increased disability), increased the risk of belonging to the PTSD with recovery group and chronic PTSD group by 1.18 and 1.17 times respectively (PTSD with recovery: [OR & (95%CI) = 1.18 (1.13, 1.23)]; chronic PTSD: [OR & (95%CI) = 1.17 (1.12, 1.22)].

Contrary to this, a one unit increase in mental quality of life score (improved mental quality of life), decreased the risk of belonging to the PTSD with recovery group and chronic PTSD group by 5% and 8% respectively (PTSD with recovery: [OR & (95%CI) = 0.95 (0.92, 0.98)]; chronic PTSD: [OR & (95%CI) = 0.92 (0.89, 0.95)]. A one unit increase in social support also decreased the risk of belonging to the PTSD vulnerable group by 20% (PTSD vulnerable: [OR & (95%CI) = 0.80 (0.72, 0.89)] compared to the resilient trajectory group (see Table 4).

## Mediation analysis of factors associated with PTSD trajectories using the Karlson-Holm-Breen (KHB) method

In the mediation analyses for PTSD trajectory group membership, perceived traumatic childbirth, was defined as the key independent variable, and depression, anxiety and WHODAS 2.0 total score as mediators. Mental quality of life score, fear of childbirth score, psychological violence, relational problems, income instability, death of loved one, mode of delivery, gravidity, parity, body mass index (BMI), monthly expenditure were included as controls.

Similarly, fear of childbirth, health risk and WHODAS 2.0 total score were also defined as key independent variables, and depression and anxiety as mediators. Social support, mode of delivery and health risk were included as controls for the key independent variable of fear of child birth. Also, for the health risk key independent variable, fear of childbirth, social support and mode of delivery were included as controls. For the WHODAS 2.0 total score key independent variable, perceived traumatic childbirth, mental quality of life score, fear of childbirth score, psychological violence, relational problems, income instability, death of loved one, mode of delivery, gravidity, parity, body mass index (BMI) and monthly expenditure were included as controls. Table 5 reports the results of the decomposition.

The results showed that the total effect of perceived traumatic childbirth on chronic PTSD trajectory group membership, 4.76 (95% CI = 3.20, 6.32), could be decomposed into a direct effect, 2.82 (95% CI = 1.37, 4.27), and indirect effect, 1.94 (95% CI = 1.25, 2.64). In relative terms, the indirect effect constituted 40.79% of the total effect. Breaking down the indirect effect into its components, it was found that a substantial part of the indirect effect was via depression (16.23%), WHODAS 2.0 total score (13.67%) and anxiety (10.89%) while controlling for other confounders.

The total effect of fear of childbirth on the chronic PTSD trajectory group membership, 0.06 (95% CI = 0.04, 0.08), could be decomposed into a direct effect, 0.03 (95% CI = 0.01, 0.05), and indirect effect, 0.03 (95% CI = 0.02, 0.05). In relative terms, the indirect effect constituted more than half of (56.93%) of the total effect. Breaking down the indirect effect into its components, it was found that a substantial part of the indirect effect was via depression (32.47%) and anxiety (24.46%).

The total and direct effects of WHODAS 2.0 total score on the chronic PTSD trajectory group membership were 0.23 (95% CI = 0.18, 0.28) and 0.16 (95% CI = 0.11, 0.20), and the indirect effect of WHODAS 2.0 total score was 0.07 (95% CI = 0.05, 0.10) with 32.58% of the total effect being mediated. Breaking down the indirect effect into its components, it was found that a substantial part of the indirect effect was via depression (16.52%) and anxiety (16.06%).

The total and direct effects of stressful life event of health risk on the PTSD with recovery trajectory group membership were 2.88 (95% CI = 1.70, 4.06) and 1.32 (95% CI = 0.16, 2.47), and indirect effect, 1.57 (95% CI = 0.64, 2.50). In relative terms, the indirect effect constituted more than half of (54.36%) the total effect. Breaking down the indirect effect into its components, it was found that a substantial part of the indirect effect was via anxiety (35.35%) and depression (19.01%) (see Table 5).

## Discussion

By using a group-based trajectory modeling of the PCL-5 longitudinal data, we have identified four distinct trajectory groups with different longitudinal patterns. The first group, which consists of mothers having zero mean of PTSD symptoms (resilient group), exhibited consistently (stable) no PTSD symptom patterns throughout the follow up period. Mothers in the PTSD vulnerable group (< the PCL-5 cut-off level which is 33) exhibited decreasing patterns (recovery) over the follow up period. Similarly, the third trajectory, labeled as PTSD with recovery group (above the PCL-5 cut-off level which is 33) exhibited decreasing patterns (recovery) over the follow up period. However, the fourth trajectory of the PTSD symptoms labeled as chronic PTSD group, exhibited relatively stable patterns of higher PTSD symptom scores (above the PCL-5 cut-off level which is 33) throughout the follow up period.

In this study, depression and anxiety were found to be independent risk factors for women to belong to the PTSD vulnerable, PTSD with recovery and chronic PTSD trajectory groups

**Table 5. Decomposition of the Total Effect of statistically significant key independent variables on PTSD trajectories into Direct and Indirect Effects using the KHB method among postpartum women, Northwest Ethiopia, 2021.**

| Key variables | PTSD trajectories (reference group: resilient with mean of zero) | | | | | | | | |
| --- | --- | --- | --- | --- | --- | --- | --- | --- | --- |
| | Vulnerable vs resilient group | | | PTSD with recovery vs resilient group | | | Chronic PTSD vs resilient group | | |
| | Coeff (SE) | (95%CI) | Mediation Percentage | Coeff (SE) | (95%CI) | Mediation Percentage | Coeff (SE) | (95%CI) | Mediation Percentage |
| **Fear of childbirth score** | | | | | | | | | |
| Total effect | 0.03(0.01) | 0.02, 0.04 | 100% | 0.07(0.01) | 0.04, 0.09 | 100% | 0.06 (0.01) | 0.04, 0.08 | 100% |
| Direct effect | 0.01(0.01) | 0.001, 0.02 | 39.06% | 0.04(0.01) | 0.01, 0.05 | 51.89% | 0.03(0.01) | 0.01, 0.05 | 43.07% |
| Indirect effect (sum) | 0.02 (0.004) | 0.01, 0.03 | 60.94% | 0.03 (0.01) | 0.02, 0.04 | 48.11% | 0.03(0.01) | 0.02, 0.05 | 56.93% |
| Via depression | 0.01 (0.003) | 0.001,0.012 | 24.30% | 0.01 (0.005) | 0.01, 0.03 | 22.99% | 0.02(0.01) | 0.005,0.02 | 32.47% |
| Via anxiety | 0.01 (0.003) | 0.004,0.02 | 36.64% | 0.02(0.01) | 0.01, 0.03 | 25.13% | 0.01 (0.005) | | 24.46% |
| **WHODAS 2.0 total score** | | | | | | | | | |
| Total effect | 0.12 (0.01) | 0.10, 0.15 | 100% | 0.24 (0.03) | 0.18, 0.29 | 100% | 0.23(0.03) | 0.18, 0.28 | 100% |
| Direct effect | 0.09 (0.01) | 0.07, 0.12 | 72.71% | 0.17(0.02) | 0.12, 0.21 | 69.45% | 0.15(0.02) | 0.11, 0.20 | 67.42% |
| Indirect effect (sum) | 0.03 (0.01) | 0.02, 0.05 | 27.29% | 0.07(0.01) | 0.05, 0.10 | 30.55% | 0.08 (0.01) | 0.05, 0.10 | 32.58% |
| Via depression | 0.01(0.01) | -0.003,0.02 | 7.35% | 0.03(0.01) | 0.01, 0.05 | 11.09% | 0.04(0.01) | 0.02, 0.06 | 16.52%) |
| Via anxiety | 0.02(0.01) | 0.01, 0.04 | 19.94% | 0.04(0.01) | 0.02, 0.07 | 19.46% | 0.04(0.01) | 0.01, 0.06 | 16.06% |
| **Health Risk: Yes** | | | | | | | | | |
| Total effect | 2.10(0.32) | 1.48, 2.73 | 100% | 2.88(0.60) | 1.70, 4.06 | 100% | | | |
| Direct effect | 1.20(0.32) | 0.58, 1.83 | 57.15% | 1.32(0.59) | 0.16, 2.47 | 45.64% | | | |
| Indirect effect (sum) | 0.90(0.27) | 0.37, 1.44 | 42.85% | 1.57(0.47) | 0.64, 2.50 | 54.36% | | | |
| Via depression | 0.25(0.13) | -0.01, 0.51 | 12.02% | 0.55(0.26) | 0.04, 1.05 | 19.01% | | | |
| Via anxiety | 0.65(0.21) | 0.23, 1.07 | 30.83% | 1.02(0.35) | 0.34, 1.70 | 35.35% | | | |
| **Traumatic birth: Yes** | | | | | | | | | |
| Total effect | | | | 3.43(0.68) | 2.10,4.75 | 100% | 4.76(0.80) | 3.20, 6.32 | 100% |
| Direct effect | | | | 1.56(0.63) | 0.33, 2.78 | 45.42% | 2.82 (0.74) | 1.37, 4.27 | 59.21% |
| Indirect effect (sum) | | | | 1.87(0.35) | 1.19, 2.55 | 54.58% | 1.94 (0.36) | 1.25, 2.64 | 40.79% |
| Via depression | | | | 0.53(0.23) | 0.08, 0.99 | 15.57% | 0.77 (0.26) | 0.27, 1.27 | 16.23% |
| Via anxiety | | | | 0.65(0.23) | 0.02, 1.09 | 18.87% | 0.52(0.21) | 0.10, 0.94 | 10.89% |
| Via WHODAS 2.0 total score | | | | 0.69(0.18) | 0.34, 1.04 | 20.14% | 0.65 (0.17) | 0.31, 0.99 | 13.67% |

relative to the resilient trajectory group. Symptoms of depression and anxiety not only were strongly related to the PTSD trajectory group membership directly, but also mediated much of the effect of the subjective traumatic birth, fear of childbirth and stressful life event of health risk on post-traumatic stress disorder trajectory group membership.

Similar findings have been reported in previous literatures that higher anxiety and depression symptom scores increased the risk of developing PTSD symptoms [3,11,75,76]. This confirms earlier findings that depression and anxiety are large component of PTSD and are highly

comorbid with PTSD after birth [11,75]. The finding that comorbid affective symptoms contribute to the persistence of birth related PTSD in this study is consistent with the finding of another study, suggesting that chronicity of PTSD may be maintained by psychological comorbidity [1].

The possible explanation for increased risk of belonging to the PTSD vulnerable, PTSD with recovery and chronic PTSD trajectory groups as a result of depressive symptoms, might be due to the negative affect and anhedonia (lack of pleasure) by depressive symptoms, which in turn increases the risk for PTSD symptoms [75]. Other explanation might be due to depressive symptoms' adverse impact on the self, which in turn increases the risk for PTSD as it has been evidenced that negative cognitions regarding the self were prospectively associated with an increase in PTSD symptoms after childbirth [75,77]. Moreover, it might be due to depressive symptoms' effect on impaired motivation for fear extinction [75]. However, other studies have reported an inverse direction of relationships that PTSD symptoms leads to depressive symptoms [78,79]. Given these diverse results, further research is needed to explain the directionality of the relationships between postpartum depression, anxiety and PTSD.

The increased risk of belonging to the PTSD vulnerable, PTSD with recovery and chronic PTSD trajectory groups as a result of anxiety symptoms might be due to childbirth related negative emotions which could overwhelm the mother and induce dissociative symptoms that interfere with the integration of traumatic memories [3]. Such emotional reactions likely indicate that birth was appraised as threatening and difficult to control which is confirmed by the result of this study in which 39.5% of mothers perceived childbirth as traumatic.

While controlling for other confounders, subjective traumatic birth was also found to be independent risk factor for women to belong to the PTSD vulnerable, PTSD with recovery and chronic PTSD trajectory groups relative to the resilient trajectory group. As for the mediation analyses, WHODAS 2.0 total score, depression and anxiety symptoms also explained in part how subjective traumatic birth experience increased the risk of belonging to the recovered and chronic PTSD trajectory. This implies that subjective traumatic birth experience intensifies symptoms of depression and anxiety and thereby predict post-traumatic stress symptoms. Our results support earlier studies highlighting the particular importance of subjective traumatic childbirth experience as it is an independent contributor for developing post-traumatic stress disorder symptoms [6,37].

Subjective childbirth perceptions, or how a woman views or interprets her childbirth experience, are often characterized by perceived experiences of the birth such as feelings of control, pain, distress, fear for self and baby and support from medical staff, partners, or other support persons which are subsequently related to posttraumatic stress disorder symptoms [6,11,37,80]. It has been suggested that subjective traumatic childbirth perception is strongly associated with postpartum posttraumatic stress disorder symptoms [12]. Severity of pain, as perceived by the woman, was also found to be associated with increased levels of postpartum posttraumatic stress disorder symptoms [11].

Consistent with the finding of this study, previous literatures have also reported fear of childbirth as a risk factor for developing postpartum PTSD and is strongly associated with traumatic subjective experiences during childbirth [6,37]. The finding that the association between fear of childbirth and belonging to the vulnerable, recovery and chronic PTSD trajectory was to a major part mediated by depression and anxiety in the mediational analyses can thus explain why fear of childbirth results in trauma responses. High levels of fear of child birth was also found to be associated with an increased risk of belonging to the recovered and chronic PTSD trajectory group in another study [1]. This might be due to higher level of incongruence between women expectations about birth and actual experiences of childbirth. When expectations about birth are not met with actual experiences of childbirth which is

more fearful than expected, a woman may feel traumatized and develop PTSD symptoms [1]. Therefore, fear of childbirth and subjective traumatic birth presents a very important point of intervention that if addressed, could possibly help to attenuate postpartum PTSD which is a negative outcome of childbirth.

Interpersonal relational problems of stressful life event also found to be risk factor for women to belong to the recovery and chronic PTSD trajectory groups in this study. Experiences of relational problems may be related to intimate partner violence (psychological or physical or sexual) which leads to psychological difficulties such as anxiety, vulnerability, low self-esteem and relationship dissatisfaction that, in turn, may affect the way the woman faces delivery [81].

While stressful life event of health risk increased the risk of belonging to the vulnerable and recovered PTSD trajectory group, stressful life event of death of loved one also found to be a risk factor of belonging to the PTSD vulnerable trajectory. However, some of the effect of stressful life event of health risk on belonging to the vulnerable and recovered PTSD trajectory group, was in part accounted for by depression and anxiety. Women with income instability stressful life event were also found to be more likely to belong to the recovery and chronic PTSD trajectory relative to the PTSD resilient trajectory. These findings might be due to the fact that successive or cumulative exposure to stressful or adverse life events might weaken one's coping ability, increase the stress -responsiveness and psychological vulnerability to trauma, and therefore increases the risk for PTSD [82].

Psychological intimate partner violence was also found to be risk factor for women to belong to the chronic PTSD trajectory group in the postpartum period, which is consistent with previous studies [81,83]. A study of PTSD at the third months of postpartum period, showed an independent effect of psychological aggression and physical assault during pregnancy on symptoms of the postpartum PTSD [83]. In another study, lifetime and recent intimate partner violence exposure (psychological, physical or sexual) were significantly associated with PTSD symptoms at the 7 and 13 months of postpartum [84]. After controlling for PTSD symptoms during pregnancy, residual PTSD symptoms at 7 and 13 months postpartum were found to be a function of greater lifetime history of intimate partner violence exposure [84]. These results suggest that risk for intimate partner violence extends beyond pregnancy, and its effects are exacerbated by interpersonal relational problems, as well as by lower social support.

The perceived higher social support was an independent protective factor of PTSD symptoms only for the PTSD vulnerable trajectory group in this study. Receiving social support is a powerful predictor for postpartum maternal health since positive social environment and support in the form of provision of resources and assurance can act as a stress buffering mechanism, protecting the mental health of mothers [85]. Similarly, multiparity decreases the risk of belonging to vulnerable, recovered and chronic PTSD trajectory group. Similar finding was reported that multiparity decreases the risk of developing PTSD symptoms following childbirth [11]. In a systematic review, it has been also reported that nulliparity to be a predisposing factor for PTSD which is in line with our study finding [25]. Nulliparity is among the vulnerability factors that were proposed to interact with birth events to determine appraisal of birth as traumatic, and subsequent traumatic stress responses [11].

In our study, while increased mental quality of life score was protective of belonging to the vulnerable, recovery and chronic PTSD trajectory, higher WHODAS 2.0 total score (increased functional impairment) was found to be a risk factor for the vulnerable, recovery and chronic PTSD trajectory group membership. As for the mediation analyses, some of the effect of WHODAS 2.0 total score on PTSD trajectory group membership, was in part accounted for by depression and anxiety. In one study, it is reported that women with PTSD symptoms had a

worse quality of life at 4–6 weeks of postpartum [86] and in another study, women with PTSD symptoms reported that the symptoms adversely affected aspects of their daily functioning [87]. Yet the reverse causal direction is conceivable, too as indicated by the finding of this study. Given these diverse results, further longitudinal researchers are needed to explain the directionality of the relationships between functional impairment and PTSD symptoms during the postpartum period. Other evidences also showed an inverse relationship of anxiety and depressive symptoms with maternal functioning [85,88,89].

## Strength and limitation of the study

Strength of this study is the use of group-based trajectory analysis to identify subgroups of longitudinal trajectories of PTSD symptoms. Group based trajectory modeling enables the identification of the distinct underlying trajectories and their predictors. Understanding these distinct trajectory subgroups and the risk factors associated with each trajectory can help to provide prognostic information for mothers and to inform the design of targeted risk factors prevention for women in the postpartum period. In addition, this study, used the Karlson-Holm-Breen (KHB) method which is appropriate for mediation analyses in nonlinear models. This method allows for the calculation of the mediated percentage, which is interpreted as the percentage of the main association that can be explained by the mediator and to include other confounding variables (as concomitants) into the models to control the decomposition of any potential confounding factors.

However, this study was not without limitations. First, antenatal factors like depression and anxiety during pregnancy and prior PTSD which may influence the prevalence and trajectories of PTSD symptoms in the postpartum period were not included in the study. Thus, it would have been better to include antenatal depression, anxiety and prior PTSD as a confounding variable. Therefore, further studies are required to clarify this issue. Second, self-report questionnaires rather than clinical interviews were used to assess PTSD symptoms which might inflate the prevalence rates. Additionally, in Bayesian analysis there is no correct way to choose a prior and it does not tell us how to select a prior. As a result, if we do not proceed with caution, a misleading result could be generated. However, to minimize this limitation, a priori we hypothesized the labeling and maximum number of trajectory groups based on the suggestion of previous research works on PTSD symptom trajectories in postpartum women.

## Conclusion and recommendation

Four distinct trajectories of postpartum posttraumatic stress disorder symptoms were identified. Perceived traumatic childbirth, fear of childbirth, depression, anxiety, psychological violence, higher WHODAS 2.0 total score, multigravidity, stressful life events of health risk, relational problems and income instability were found to be predictors of PTSD with recovery and chronic PTSD trajectory group membership. Depression and anxiety not only were strongly related to trajectories of PTSD symptoms directly but also mediated much of the effect of the other factors on trajectories of PTSD symptoms. In contrast, multiparity and higher mental quality of life scores were protective of belonging to the PTSD with recovery and chronic PTSD trajectory group membership. Therefore, postnatal screening and treatment of depression and anxiety may contribute to decrease PTSD symptoms of women in the postpartum period. Providing adequate information about birth procedures and response to mothers' needs during childbirth and training of health care providers to be mindful of factors that contribute to negative appraisals of childbirth are essential to reduce fear of childbirth and traumatic childbirth so as to prevent PTSD symptoms in the postpartum period.

## Supporting information

**S1 Dataset. PTSD trajectory dataset.**
(DTA)

## Acknowledgments

The authors would like to acknowledge the heads of Debre Tabor Hospital, Addis Zemen Hospital, Estie Hospital and Nefas Mewcha Hospital for their cooperation on the data collection of this study. The authors are also grateful to the study participants for their dedicated time and volunteer participation.

## Declarations

**Ethics approval and consent to participate.** Ethical approval was obtained from Institutional Review Board of Bahir Dar University (Reference number: 00225/2020).

## Author Contributions

**Conceptualization:** Marelign Tilahun Malaju.

**Data curation:** Marelign Tilahun Malaju.

**Formal analysis:** Marelign Tilahun Malaju.

**Investigation:** Marelign Tilahun Malaju.

**Methodology:** Marelign Tilahun Malaju, Getu Degu Alene, Telake Azale Bisetegn.

**Project administration:** Marelign Tilahun Malaju.

**Resources:** Marelign Tilahun Malaju, Getu Degu Alene.

**Software:** Marelign Tilahun Malaju.

**Supervision:** Marelign Tilahun Malaju, Getu Degu Alene, Telake Azale Bisetegn.

**Writing – original draft:** Marelign Tilahun Malaju.

**Writing – review & editing:** Marelign Tilahun Malaju, Getu Degu Alene, Telake Azale Bisetegn.

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
