## [Decision Letter · Decision Letter 0]

4 Mar 2022

PONE-D-22-01081Longitudinal mediation analysis of the factors associated with trajectories of posttraumatic stress disorder symptoms among postpartum women in Northwest Ethiopia: Application of the Karlson-Holm-Breen (KHB) methodPLOS ONE

Dear Dr. Malaju,

Thank you for submitting your manuscript to PLOS ONE. After careful consideration, we feel that it has merit but does not fully meet PLOS ONE’s publication criteria as it currently stands. Therefore, we invite you to submit a revised version of the manuscript that addresses the points raised during the review process.

The reviewers were enthusiastic about your manuscript. However, there were some modifications that were suggested to improve the manuscript. Please see their comments and revise your manuscript appropriately.

We look forward to receiving your revised manuscript.

Kind regards,

Samuel Wilkinson, MD

Academic Editor

PLOS ONE

Journal Requirements:

2 .Please note that in order to use the direct billing option the corresponding author must be affiliated with the chosen institute. Please either amend your manuscript to change the affiliation or corresponding author, or email us at plosone@plos.org with a request to remove this option.

4.We note that you have stated that you will provide repository information for your data at acceptance. Should your manuscript be accepted for publication, we will hold it until you provide the relevant accession numbers or DOIs necessary to access your data. If you wish to make changes to your Data Availability statement, please describe these changes in your cover letter and we will update your Data Availability statement to reflect the information you provide.

6. We noticed you have some minor occurrence of overlapping text with the following previous publication(s), which needs to be addressed:

- https://openaccess.city.ac.uk/id/eprint/19244/1/Running%20head%20Trajectories%20of%20birth-related%20PTSD.pdf

- https://journals.sagepub.com/doi/10.1177/1359105318787018

The text that needs to be addressed involves the Introduction.

In your revision ensure you cite all your sources (including your own works), and quote or rephrase any duplicated text outside the methods section. Further consideration is dependent on these concerns being addressed.

Additional Editor Comments:

Dear authors -

The reviewers were enthusiastic about your manuscript. However, there were some modifications that were suggested to improve the manuscript. Please see their comments and revise your manuscript appropriately.

Reviewers' comments:

Reviewer's Responses to Questions

**Comments to the Author**

1. Is the manuscript technically sound, and do the data support the conclusions?

Reviewer #1: Yes

Reviewer #2: Yes

2. Has the statistical analysis been performed appropriately and rigorously? 

Reviewer #1: Yes

Reviewer #2: I Don't Know

3. Have the authors made all data underlying the findings in their manuscript fully available?

Reviewer #1: Yes

Reviewer #2: No

4. Is the manuscript presented in an intelligible fashion and written in standard English?

Reviewer #1: Yes

Reviewer #2: Yes

5. Review Comments to the Author

Reviewer #1: Thank you for the opportunity to review this manuscript! I find the results obtained extremely valuable and important. The procedure and the results are clearly described. Women with still birth were included in the sample. I suggest to mention this group in the results section, as antenatal loss is known to be a strong predictor of PTSD. And I have a question about the age variable. The inclusion criterion was 15 y.o. Were there any differences in the teenage mothers group regarding pospartum PTSD risks?

Reviewer #2: The authors investigated the trajectory of PTSD symptoms in Ethiopian women postpartum. They also conducted a mediation analysis to determine the factors associated with each trajectory. They found 4 trajectories: 1) no symptoms, 2) subthreshold, 3) transient symptoms, and 4) persistent symptoms.

Various factors were associated with the three trajectories comprising PTSD symptoms compared to symptoms, including high levels of depression and anxiety.

The manuscript is clear and comprehensive but very lengthy at times. The methods, results and discussion sections could be shortened. Some details can be moved to Online Supplements. Otherwise, the research question is of relevance to the field. While I cannot comment on the statistical methods, overall the approach seems adequate, the results are informative and the conclusion are primarily reasonable.

6. PLOS authors have the option to publish the peer review history of their article (what does this mean?). If published, this will include your full peer review and any attached files.

Reviewer #1: **Yes: **Vera Yakupova

Reviewer #2: No

---

## [Author Response · Author response to Decision Letter 0]

8 Mar 2022

Author’s response for reviewers’ comments

Response for Editor(s)' Comments:

Comments:

Journal Requirements:

Response:

The manuscript is formatted in line with the journal requirement. 

Now the manuscript meets PLOS ONE's style requirements, including those for file naming (see the clean copy and marked copy of the revised document)

Comments:

2. Please note that in order to use the direct billing option the corresponding author must be affiliated with the chosen institute. Please either amend your manuscript to change the affiliation or corresponding author, or email us at plosone@plos.org with a request to remove this option.

Response: Please remove the direct billing option. I have sent an email to plosone@plos.org with a request to remove this option. I was not aware of it. 

Comments:

Response: 

Now, I have uploaded my study’s minimal underlying data set as Supporting Information files.

Comments:

4.We note that you have stated that you will provide repository information for your data at acceptance. Should your manuscript be accepted for publication, we will hold it until you provide the relevant accession numbers or DOIs necessary to access your data. If you wish to make changes to your Data Availability statement, please describe these changes in your cover letter and we will update your Data Availability statement to reflect the information you provide.

 Response:

Now, I have already uploaded the minimal dataset as supporting information. Previously I understand the word “repository information for my data at acceptance” as to mean that uploading the supporting information of data file up on acceptance. So, I didn’t understand it clearly. So, please update my Data Availability statement in line with providing minimal dataset as supporting information. 

Comments:

Response:

Now, I have updated my references list in line with the journal’s requirement. No retracted papers have been used

Comments:

6. We noticed you have some minor occurrence of overlapping text with the following previous publication(s), which needs to be addressed:

- https://openaccess.city.ac.uk/id/eprint/19244/1/Running%20head%20Trajectories%20of%20birth-related%20PTSD.pdf

- https://journals.sagepub.com/doi/10.1177/1359105318787018

The text that needs to be addressed involves the Introduction.

In your revision ensure you cite all your sources (including your own works), and quote or rephrase any duplicated text outside the methods section. Further consideration is dependent on these concerns being addressed.

Response:

Now, some of the overlapping texts in few paragraphs are rephrased and all sources have been cited accordingly in the introduction section. 

Comments:

Additional Editor Comments:

Dear authors -

The reviewers were enthusiastic about your manuscript. However, there were some modifications that were suggested to improve the manuscript. Please see their comments and revise your manuscript appropriately.

Response:

See my response for reviewers’ comment bellow.

Response for reviewers’ comments 

Comments:

Reviewers' comments:

Reviewer's Responses to Questions

Comments to the Author

1. Is the manuscript technically sound, and do the data support the conclusions?

Reviewer #1: Yes

Reviewer #2: Yes

Response: It is ok

2. Has the statistical analysis been performed appropriately and rigorously?

Reviewer #1: Yes

Reviewer #2: I Don't Know

Response: It is shown in the methods and results section. 

3. Have the authors made all data underlying the findings in their manuscript fully available?

Reviewer #1: Yes

Reviewer #2: No

Response: Now, I have uploaded the minimal dataset.

4. Is the manuscript presented in an intelligible fashion and written in standard English?

Reviewer #1: Yes

Reviewer #2: Yes

Response: It is ok.

5. Review Comments to the Author

 Response for 1st reviewer comments 

Comments:

Reviewer #1: Thank you for the opportunity to review this manuscript! I find the results obtained extremely valuable and important. The procedure and the results are clearly described. Women with still birth were included in the sample. I suggest to mention this group in the results section, as antenatal loss is known to be a strong predictor of PTSD. And I have a question about the age variable. The inclusion criterion was 15 y.o. Were there any differences in the teenage mothers group regarding pospartum PTSD risks?

 Response

Now, I have included the result for teenage mothers’ group in Table 1 and still birth in Table 3. But, since the number of events were rare, there was no significant association with the PTSD trajectory group. Please, see the number of events(frequencies) for these variables in the specified tables, to appreciate how rare were these variables. 

Response for 1st reviewer comments 

Comments:

Reviewer #2: The authors investigated the trajectory of PTSD symptoms in Ethiopian women postpartum. They also conducted a mediation analysis to determine the factors associated with each trajectory. They found 4 trajectories: 1) no symptoms, 2) subthreshold, 3) transient symptoms, and 4) persistent symptoms.

Various factors were associated with the three trajectories comprising PTSD symptoms compared to symptoms, including high levels of depression and anxiety.

The manuscript is clear and comprehensive but very lengthy at times. The methods, results and discussion sections could be shortened. Some details can be moved to Online Supplements. Otherwise, the research question is of relevance to the field. While I cannot comment on the statistical methods, over all the approach seems adequate, the results are informative and the conclusion are primarily reasonable.

Response:

The improve the length of the manuscript, the following portions were removed from the specified sections of the manuscript (see below):

Abstract section (result subsection): The following paragraphs (sentences) were removed.

Prevalence rates of PTSD symptoms were 9.7%, 6.8% and 3.5% at the 6th, 12th and 18th week of postpartum respectively.

Method section (Data processing and analysis subsection): The following paragraphs (sentences) were removed.

Statistically, group-based trajectory models use maximum likelihood estimation to estimate both the trajectory of each group (modeled as a function of time using flexible polynomials) and the expected population-level distribution of each group that creates the best fit for the observed data (64, 65). Since the number of groups and the order of the trajectory polynomials (i.e., linear, quadratic, cubic) are not actually known a priori (but must be prespecified when estimating a model), we systematically tested a series of model specifications. This was done first by varying the number of groups and then the order of the trajectory polynomials in order to select the model most optimized for fit and parsimony (64, 66).

 Posterior probabilities represent the average probability that the trajectory group each participant was assigned to is the most appropriate group selection and model fit. 

This is especially true when multiple mediators are considered in the mediation analysis

 Result section (Prevalence rates of PTSD and trajectory groups by obstetrics and psychosocial variables). The following paragraphs (sentences) were removed.

The prevalence rates of PTSD symptoms at the 6th, 12th and 18th week of postpartum period were found to be 9.7%, 6.8% and 3.5% respectively.

Result section (Predictors of PTSD (PCL-5) trajectory group membership). The following paragraphs (sentences) were removed. 

In addition, participants with higher depression and anxiety score were more likely to belong to the PTSD with recovery and chronic PTSD trajectory group. Finally, mothers with stressful life events of relational problems and income instability were also more likely to belong to the PTSD with recovery and chronic PTSD trajectory group.

Result section (Mediation analysis of factors associated with PTSD trajectories using the Karlson-Holm-Breen (KHB) method). The following paragraphs (sentences) were removed. 

Similarly, the total effect of perceived traumatic childbirth on the PTSD with recovery trajectory group membership, 3.43 (95% CI= 2.10,4.75), could be decomposed into a direct effect, 1.56 (95% CI= 0.33, 2.78), and indirect effect, 1.87 (95% CI= 1.19, 2.55). In relative terms, the indirect effect constituted more than half of (54.58%) of the total effect. Breaking down the indirect effect into its components, it was found that a substantial part of the indirect effect was via WHODAS 2.0 total score (20.14%) anxiety (18.87%) and depression (15.57%) while controlling for other confounders. All mediating effects were statistically significant. 

The total effect of fear of childbirth on the PTSD with recovery trajectory group membership, 0.07 (95% CI= 0.04, 0.09), could be also decomposed into a direct effect, 0.04 (95% CI= 0.01, 0.05), and indirect effect, 0.03 (95% CI= 0.02, 0.04). In relative terms, the indirect effect constituted 48.11% of the total effect. Breaking down the indirect effect into its components, it was found that a substantial part of the indirect effect was via anxiety (25.13%) and depression (22.99%). Likewise, the total effect of fear of childbirth on the PTSD vulnerable trajectory group membership, 0.03 (95% CI= 0.02, 0.04), could be decomposed into a direct effect, 0.01 (95% CI= 0.001, 0.02), and indirect effect, 0.02 (95% CI= 0.01, 0.03). In relative terms, the indirect effect constituted more than half of (60.94%) of the total effect. Breaking down the indirect effect into its components, it was found that a substantial part of the indirect effect was via anxiety (36.64%) and depression (24.30%). All mediating effects were statistically significant. 

The total and direct effects of WHODAS 2.0 total score on the PTSD with recovery trajectory group membership were 0.24 (95% CI= 0.18, 0.29) and 0.17 (95% CI= 0.12, 0.21), and indirect effect, 0.07 (95% CI= 0.05, 0.10). In relative terms, the indirect effect constituted 30.55% of the total effect. Breaking down the indirect effect into its components, it was found that a substantial part of the indirect effect was via anxiety (19.46%) and depression (11.09%). 

The total and direct effects of WHODAS 2.0 total score on the PTSD vulnerable trajectory group membership were 0.12 (95% CI= 0.10, 0.15) and 0.09 (95% CI= 0.07, 0.12), and indirect effect, 0.03 (95% CI= 0.02, 0.05). In relative terms, the indirect effect constituted 27.29% of the total effect. Breaking down the indirect effect into its components, it was found that a substantial part of the indirect effect was via anxiety (19.94%) and depression (7.35%). 

The total and direct effects of stressful life event of health risk on the PTSD vulnerable trajectory group membership were 2.10 (95% CI= 1.48, 2.73) and 1.20 (95% CI= 0.58, 1.83), and indirect effect, 0.90 (95% CI= 0.37, 1.44). In relative terms, the indirect effect constituted 42.85% of the total effect. Breaking down the indirect effect into its components, it was found that a substantial part of the indirect effect was via anxiety (30.83%) and depression (12.02%) 

Discussion section. The following paragraphs (sentences) were removed. 

In this study, the prevalence rates of postpartum PTSD symptoms, were found to be 9.7%, 6.8% and 3.5% at the 6th, 12th and 18th week of postpartum respectively. This is consistent with a study conducted among perinatal women in Turkey(76). However, 9.7% and 6.8% prevalence of PTSD in our study, is considerably higher than the finding of a systematic review which reported 4% postpartum PTSD(13). The postpartum PTSD symptoms might be due to difficult childbirth experiences triggering new onset of PTSD or exacerbating pre-existing PTSD symptoms. Other possible explanation could be, a difficult birth might retrigger PTSD which was experienced in early life. The prevalence of 3.5% PTSD at the 18th week of postpartum, suggests that PTSD after childbirth might be chronic and indicates that some women with negative birth experiences maintained their negative perception of childbirth and their symptoms until 18th week of postpartum. 

Conclusion section: The following paragraphs (sentences) were removed.

Prevalence rates of PTSD symptoms were 9.7%, 6.8% and 3.5% at the 6th, 12th and 18th week of postpartum respectively. 

I thank you very much!

---

## [Editor Report · Decision Letter 1]

21 Mar 2022

Longitudinal mediation analysis of the factors associated with trajectories of posttraumatic stress disorder symptoms among postpartum women in Northwest Ethiopia: Application of the Karlson-Holm-Breen (KHB) method

PONE-D-22-01081R1

Dear Dr. Malaju,

We’re pleased to inform you that your manuscript has been judged scientifically suitable for publication and will be formally accepted for publication once it meets all outstanding technical requirements.

Kind regards,

Samuel Wilkinson, MD

Academic Editor

PLOS ONE
---

## [Editor Report · Acceptance letter]

25 Mar 2022

PONE-D-22-01081R1 

Longitudinal mediation analysis of the factors associated with trajectories of posttraumatic stress disorder symptoms among postpartum women in Northwest Ethiopia: Application of the Karlson-Holm-Breen (KHB) method 

Dear Dr. Malaju:

I'm pleased to inform you that your manuscript has been deemed suitable for publication in PLOS ONE. Congratulations! Your manuscript is now with our production department. 

Kind regards, 

on behalf of

Dr. Samuel Wilkinson 

Academic Editor

PLOS ONE